# Antithrombotics from the Sea: Polysaccharides and Beyond

**DOI:** 10.3390/md17030170

**Published:** 2019-03-16

**Authors:** Francisca Carvalhal, Ricardo R. Cristelo, Diana I. S. P. Resende, Madalena M. M. Pinto, Emília Sousa, Marta Correia-da-Silva

**Affiliations:** 1LQOF—Laboratório de Química Orgânica e Farmacêutica, Departamento de Ciências Químicas, Faculdade de Farmácia, Universidade do Porto, Rua de Jorge Viterbo Ferreira, 228, 4050-313 Porto, Portugal; francisca.carvalhal@gmail.com (F.C.); ricardorc@live.com.pt (R.R.C.); dresende@ff.up.pt (D.I.S.P.R.); madalena@ff.up.pt (M.M.M.P.); m_correiadasilva@ff.up.pt (M.C.-d.-S.); 2CIIMAR—Centro Interdisciplinar de Investigação Marinha e Ambiental, Terminal de Cruzeiros do Porto de Leixões, Av. General Norton de Matos S/N, 4450-208 Matosinhos, Portugal

**Keywords:** marine-derived, polysaccharide, anticoagulant, antithrombotic, antiplatelet, glycosaminoglycans, sulfated fucans, sulfated galactans

## Abstract

Marine organisms exhibit some advantages as a renewable source of potential drugs, far beyond chemotherapics. Particularly, the number of marine natural products with antithrombotic activity has increased in the last few years, and reports show a wide diversity in scaffolds, beyond the polysaccharide framework. While there are several reviews highlighting the anticoagulant and antithrombotic activities of marine-derived sulfated polysaccharides, reports including other molecules are sparse. Therefore, the present paper provides an update of the recent progress in marine-derived sulfated polysaccharides and quotes other scaffolds that are being considered for investigation due to their antithrombotic effect.

## 1. Introduction

Over the last 15 years, according to the World Health Organization (WHO), the world’s leading causes of death are ischemic heart disease and stroke, which constitute the number one and two of the top 10 global causes of death and accounted for a combined 15.2 million deaths in 2016 (WHO 2018) [1,2]. To treat ischemic heart disease and stroke, antiplatelet agents and, in certain circumstances, anticoagulants, are used. Heparin (HP) was the first clinically used anticoagulant, and after more than 100 years of its discovery (1916), it remains the most used [3], even though its use is limited to parenteral administration (mainly because of their highly negative charge and large molecular weight) and the new orally active anticoagulants lack the multiple activities of HP, which is thought to be involved beyond the coagulation cascade [4], with antimetastatic [5] and anti-inflammatory activities [6]. The search for orally active and multitarget small molecules as new antithrombotic drugs is an attractive approach to overcome the limitations of current drugs used in therapy [7].

Significant interest has been established in anticancer and antimicrobial compounds isolated from marine sources due to their unique and unusual chemical structures, as well as pharmacological properties [8]. In recent years, other therapeutic areas are being populated by marine bioactive secondary metabolites, cardiovascular being among these areas of unmet medical needs. The most abundant marine sources of new antithrombotic compounds are marine algae and invertebrates, producing both macromolecules, such as polysaccharides, and small molecules (e.g., peptides, terpenes, alkaloids, polyphenols, steroids, and polyketides). Although over the last few decades, polysaccharides have been identified as the most therapeutically explored metabolites and suppliers of new antithrombotic agents; in fact, the less explored small molecules have proven to also be an excellent starting point for the development of new and orally effective drugs.

While comprehensive and updated reviews can be found in the literature regarding marine sulfated polysaccharides with antithrombotic activities [9,10,11,12,13,14,15,16,17,18,19,20,21,22,23,24,25,26,27,28,29,30,31], reports of marine antithrombotic small molecules are still sparse in the literature. Therefore, we performed a run-up review of some of the major representatives and newest examples of bioactive polysaccharides reported in the last five years, and a systematic compilation of small molecules isolated from marine organisms with antithrombotic activities.

## 2. Methods Used to Evaluate Antithrombotic Activities of Marine Compounds

Marine-derived compounds were evaluated for their effects on coagulation and platelet aggregation. In a few cases, fibrinolytic activity was also determined.

### 2.1. Evaluation of Coagulation

#### 2.1.1. Classical Coagulation Times

The influence of compounds on the plasmatic blood coagulation was usually determined by measuring the activated partial thromboplastin time (APTT), prothrombin time (PT), and thrombin time (TT), which allow the evaluation of the ability to inhibit blood clotting through the intrinsic, extrinsic, and common pathways of the coagulation cascade, respectively (Figure 1) [32].

Usually, by venipuncture, nine parts of blood are taken into one part of 3.2% sodium citrate, to prevent blood clotting, from healthy volunteers who did not take any antithrombotic medication. The plasma obtained after centrifugation was pooled and mixed with a compound solution (usually in saline). Coagulation assays were performed using an automated coagulation analyzer and diagnostic kit, and the time that it takes to form a clot was measured in seconds [33,34,35,36,37].

#### 2.1.2. Effects on Coagulation Factors

##### Chromogenic Assays

Chromogenic assays were used for the characterization of the affinity of the tested compounds toward their target enzymes. Most often, synthetic substrates onto which a chromophore has been linked and purified coagulation proteases were used [36,38,39]. Proteases cleave the chromogenic substrate, releasing a colored compound that can be monitored using a spectrophotometric reader. When the protease was previously incubated with inhibitory marine compounds, a decrease of the chromogenic substrate cleavage was detected [36,38,39].

##### Modified Clotting Assays

In order to identify the specific factor inhibition underlying the APTT prolongation of a marine fish protein (Section 3.2.1), human plasma with specific factor deficiencies from the intrinsic pathway was used [40]. The principle of this assay is to measure to what extent a normal plasma incubated with the compound avoids the normalization of the coagulation time of the specific factor-deficient plasma, which happens when normal plasma (without the compound) is added.

##### Surface Plasmon Resonance Spectroscopy

To identify the binding affinity of several marine antithrombotic peptides to blood coagulation factors (Section 3.2.1), surface plasmon resonance sensorgrams were obtained [33,34,41]. In this assay, a ligand protein was directly immobilized on a sensor chip surface and unreacted groups were blocked with ethanolamine. After immobilization of coagulation factors on the sensor chip, the analyte was injected onto the surface of the sensor chip in HEPES-buffered saline buffer followed by dissociation. Resonance was monitored as a function of time, and the binding affinity was represented as a resonance unit or response unit in real time.

##### PAGE Analysis of Clotting Factor Inhibition

In order to identify the inhibition of clotting factor (F) XII activated (XIIa) by a marine fish protein (Section 3.2.1), a native polyacrylamide gel electrophoresis (PAGE) method was used. A mixture of the specific factor and analyte were incubated in the presence or absence of Zn^2+^. After the reaction, aliquots migrated on a 12.5% separating gel at pH 8.8 using an electrolyte buffer. Several calibration molecular weight markers were used [40].

### 2.2. Evaluation of Platelet Aggregation

Concerning antiplatelet activity, several marine compounds were incubated with platelet-rich plasma (PRP) or washed platelets (WP) or whole blood and then, platelet aggregation was triggered by adding various types and doses of agonists, such as adenosine diphosphate (ADP), collagen, thrombin, and/or platelet activating factor (PAF) (Figure 2).

The platelet aggregation was measured turbidometrically by an aggregometer (example in Reference [37]).

### 2.3. Evaluation of Fibrinolysis

Fibrin(ogen)olytic properties of eckol, dieckol, and phlorofucofuroeckol A (Section 3.2.4) were evaluated by their inhibitory activity on the main plasmin inhibitors, α_2_-macroglubin and α_2_-antiplasmin [42,43]. Plasmin is an important enzyme of the fibrinolytic system that cuts the fibrin mesh.

### 2.4. In Vivo Antithrombotic Activity

The in vivo activity of a polysaccharide from a green alga was investigated after being injected to anaesthetized rats (Section 3.1.2). Blood was taken from the abdominal aorta, and the clotting times APTT, TT, and PT were evaluated. In vivo fibrin(ogen)olytic properties of this polysaccharide were also tested in the rat blood using fibrin(ogen) degradation products, D-dimer, and plasminogen activator inhibitor-1 (PAI-1) commercial kits (Section 3.1.2) [44]. Other assays were also used in in vivo studies, as follows.

#### 2.4.1. Recovery Time from Paralysis

A pulmonary thromboembolism model induced by the intravenous injection of collagen and epinephrine agonists was used to evaluate in vivo antithrombotic effects of two marine terpenes (Section 3.2.2) [45]. In this model, analytes and a positive control were intravenously administered into the tail veins of mice. Then, a mixed solution of pulmonary thrombosis agonists was injected into the tail vein, which resulted in mouse paralysis for a period of 40 min or death. The antithrombotic efficacy was evaluated based on the recovery time from pulmonary thrombosis-induced paralysis, compared with positive control.

#### 2.4.2. Bleeding Time

In this assay, male mice were anesthetized with pentobarbital and the analyte or HP (as control) were given. The tail vein of the mouse was wounded, and the blood was absorbed into a filter paper every 1.5 s. Bleeding time can be calculated at the point in which no blood is detected on the paper (performed for plancinin in Section 3.2.1) [46]. The bleeding time (see for example urchin sulfated glycans in Section 3.1) can also be evaluated by immersing the tail in water and following that, hemoglobin is measured by spectrophotometry [47].

#### 2.4.3. Venous Thrombosis

The in vivo venous thrombosis assay used thromboplastin as the thrombogenic stimulus (done for three urchins sulfated glycans, Section 3.1). The inferior vena cava was isolated, and different doses of the test compounds were infused and allowed to circulate for 5 min. Then, the formed thrombus was removed and weighed and compared with the thrombus weight in the absence of analyte administration [47].

## 3. Marine Compounds with Antithrombotic Activity

### 3.1. Polysaccharides

Carbohydrates (also known as glycans) are the most abundant natural biomolecules, exhibiting a large structural diversity due to the multiplicity of interactions of their monosaccharide units. These building blocks can be linked to each other at various positions and with several linkages on the pyranose or furanose rings, creating branched structures and thus presenting diverse biological functions [9,48]. Among carbohydrates, polysaccharides (PS) represent some of the most abundant bioactive substances [48]. Glycosaminoglycans (GAGs) are complex, linear, and negatively charged polysaccharides present in animal tissues [49,50]. The commonest examples of GAGs found in mammals are divided into five main groups (Figure 3): HP/heparan sulfate (HS), chondroitin sulfate (CS), dermatan sulfate (DS), keratan sulfate (KS), and hyaluronan (the only GAG that is not sulfated). Depending on the source, GAGs can have distinct structures. In marine algae and invertebrates, into addition to structurally unique GAGs such as fucosylated chondroitin sulfate (FCS), DS, and HS, some GAG-resembling sulfated glycans, like sulfated fucans (SFs) and sulfated galactans (SGs), can also be found (Figure 3) [50].

Structurally, GAGs are composed of repeating units of *N*-acetylated or *N*-sulfated hexosamine (glucosamine (GlcN)) with either uronic acid or galactose. The structures of all the sulfated GAGs and of hyaluronan are represented in Figure 4. Uronic acid can be either glucuronic acid (GlcA) or iduronic acid (IdoA). For example, HP consists of alternating *N*,6-di-*O*-sulfated α-d-GlcN (GlcNS6S) and 2-sulfated α-IdoA units (IdoA2S), both 4-linked (Figure 4), while HS is composed of repeating units of β-d-GlcA and *N*-acetylated α-d-GlcN both 4-linked [50]. Although they are not within the scope of this review, the representation of the mammalian GAGs will be of use for further comparison of structural features between mammalian and marine-derived sulfated polysaccharides.

#### 3.1.1. Marine GAGs

Marine GAGs display different structures from those of common mammalian GAGs [51], with unique sulfation patterns. For example, HS, DS, and CS from marine invertebrates have equivalent amounts of uronic acid and hexosamine as HP but with distinct sulfation patterns, being considered a valid source of HP analogues [52]. Until 2014, several examples of structurally unique GAGs with anticoagulant and antithrombotic effects isolated from ascidians [53,54,55,56,57], sea urchins [58], sea cucumbers [51,59,60,61], mollusks [62], and shrimps [63,64] were described in the literature. The main feature of these new and interesting GAGs isolated from the extracellular matrices of certain marine invertebrates is the structural regularity that, contrary to mammalian GAGs, allows the development of advanced structure–anticoagulation relationship studies [17,19,51,60]. The main conclusions of these studies point out the importance of specific structural combinations, such as the 2,4-di-sulfation with α-l-Fucp units found in the branched FCS molecules from sea cucumbers or the combination of the 4-sulfation in GalNAc units with occasional 2-sulfation in IdoA, as is the case of DS from ascidians (Figure 5) [17,51].

Over the years, several antithrombotic GAGs have been isolated from marine invertebrates. Since 2014 and to the best of our knowledge, new secondary metabolites with these activities have only been reported mainly from FCS, and almost exclusively from sea cucumbers (Table 1) [10,65,66,67,68,69]. FCSs isolated from the sea cucumber *Cucumaria frondosa* (FCSc), *Thelenota ananas* (FCSt), and its depolymerized low molecular weight (LMW) fragments (dFCSc and dFCSt) containing diverse sulfated fucose branches were evaluated by Liu and co-workers for their in vivo anticoagulant and antithrombotic activities [65]. The results demonstrated that both LMW fragments had similar antithrombotic effects and bleeding side effects and also prolong-APTT, anti-FIIa, and anti-FXa activities. Generally, the LMW fragments exhibited better antithrombotic–hemorrhagic ratios than their native forms, even when compared with HP and low molecular weight heparin (LMWH) in a rat arterial thrombosis model. Furthermore, when compared to FCSt, FCSc possessed different sulfation patterns but similar antithrombotic effects. As a conclusion, the authors stated that anticoagulation and antithrombotic effects might not be affected so much by the sulfation pattern of FCS, but the molecular weight and the sulfation degree could have some influence on the obtained results [65].

Other FCS isolated from the body wall of the Pacific sea cucumber *Cucumaria japonica* inhibited platelets aggregation in vitro mediated by collagen and ristocetin but not adenosine diphosphate [66]. Regarding anticoagulant activity, FCS exhibited higher activity than that one of LMWH in the APTT assay. The effect of FCS on the activity of thrombin and FXa was also studied in the presence and in the absence of antithrombin III (ATIII). FCS showed a comparable level of activity with that of LMWH regarding experiments with thrombin, and in the case of FXa inhibition FCS was ~10-fold less active than LMWH, both in the presence of ATIII, while no activity was observed in the absence of ATIII. These results evidenced a serpin-dependent mechanism of action of FCS in the cases of thrombin and FXa [66]. Later, the same group isolated an FCS (named MM by the authors, Figure 6) from the sea cucumber *Massinium magnum*. MM was shown to contain a typical CS core with a small portion of CS fragments. Moreover, Fuc3S4S attached to *O*-3 of GlcA residues were the only type of branches found in the structure of MM, which was determined as →4)-[α-l-Fuc3S4S-(1→3)]-β-d-GlcA-(1→3)-β-d-GalNAc4S6S-(1→, whereas the minor repeating unit was →4)-[α-l-Fuc3S4S-(1→3)]-β-d-GlcA-(1→3)-β-d-GalNAc4S-(1→(Figure 6). APTT and TT tests were used to evaluate the anticoagulant activity of MM, which was revealed to be higher (two-fold delay of clot formation (2APTT) = 2.8 ± 0.1 µg/mL) than that of enoxaparin (3.9 ± 0.2 µg/mL) but lower than that of HP (1.2 ± 0.1 µg/mL). The authors also studied the effect of MM on the activity of thrombin and FXa in the presence and in the absence of ATIII, with similar results. Although MM exhibited a comparable level of thrombin inhibition to that of enoxaparin but lower than that of HP, regarding FXa inhibition, MM was ~10-fold less active than enoxaparin. Further, no activity was observed in the absence of ATIII, suggesting a serpin-dependent mechanism of action of MM in the cases of thrombin and FXa. MM did not induce platelet aggregation in PRP.

A novel FCS (FCShm) was isolated from the sea cucumber *Holothuria mexicana*, and its anticoagulant activity was evaluated using APTT, PT, and TT of plasma clotting assays [68]. FCShm displayed intrinsic anticoagulation since, compared to LMWH, it significantly prolonged the APTT and TT and barely affected PT. Furthermore, FCShm inhibited the activities of thrombin and FXa through high binding affinity to ATIII [68]. Another example that relates to the study of the anticoagulant activities of the FCS is HsG, isolated from sea cucumber *Holothuria scabra* [69]. APTT and TT assays exhibited a similar result when comparing HsG with HP, being prolonged both times [69].

Regarding structure–activity relationship (SAR) studies, there are some controversial opinions in the literature about the influence of the structure of fucosyl branches on biological activities of FCS. Although earlier papers stated the importance of 2,4-disulfation of fucose residues for anticoagulant properties [61,70], recent publications (as discussed above) showed that the molecular weight might have a stronger influence on anticoagulation and antithrombotic events than the sulfation pattern of FCS [65,67].

#### 3.1.2. Marine GAG Mimetics

The marine environment is also a rich source of structurally unique GAGs, known as GAG mimetics, such as SFs and SGs isolated from certain macroalgae (brown [31,34,71,72,73,74,75,76,77,78,79,80,81,82,83,84,85,86,87,88,89,90], red [91,92,93,94,95,96,97,98,99,100,101,102,103], and green [72,104,105,106,107,108,109,110,111,112,113,114,115,116,117,118] (Figure 7)), microalgae [119,120,121] or from invertebrates [47,58,65,122,123,124,125,126,127,128,129] like echinoderms (sea cucumber and sea urchins) or tunicates (ascidians). These interesting sulfated homopolysaccharides have been largely studied in the last years [22,27,31] and discussed in several reviews as potential pharmaceuticals of the future [14,17,130,131].

The reasons for the growing interest in these molecules consist of (1) substantially lower contamination levels of virus and/or prions, since they are exclusively extracted from marine sources [19]; (2) the unique and distinct structures of these glycans compared to the GAG structure [24,132]; (3) the mechanisms of action that, although being similar to the GAGs used in medicine, can exhibit additional or slightly different effects, which can be considered advantageous factors in the development of alternative anticoagulants [13]; (4) the fact that some SFs and SGs do not exhibit bleeding risks, contrarily to the HP therapy [133]. In general, algal polysaccharides are structurally more complex, with heterogeneous structures, compared to polysaccharides isolated from marine invertebrates that have simple, linear structures [134].

Fucoidan designates a family of sulfated polysaccharides extracted from marine brown algae (*Phaeophycophyta*) and some echinoderms (sea urchin and sea cucumber). Most of the structures found in brown algae are highly heterogeneous due to several sulfation and glycosylation sites and also the common presence of branching residues in any position, making complete structural elucidation difficult [134]. The term SFs is reserved for polysaccharides with a regular structure, containing a majority of sulfated fucose, which are often extracted from marine invertebrates [12]. The chemical structures of these SFs were found to be species-specific [122]. Since fucoidan preparations are often mixtures of structurally different polysaccharides, the structure elucidation of these is often problematic. Four fractions from a fucoidan preparation named FSA (by the authors) were isolated from the brown algae species *Sargassum aquifolium* collected from the coastal waters of Vietnam [135]. These fractions were analyzed by chemical and spectroscopic methods (nuclear magnetic resonance) and revealed the presence of three structurally different polysaccharides. HP-like anticoagulant properties of FSA fractions were characterized by determination of APTT, and some fractions showed 2APTT = 6.5 ± 0.4 µg/mL for 2.0 M when compared to enoxaparin (3.9 ± 0.4 µg/mL), even in the most sulfated fractions [135]. The same authors also reported the isolation of a mixture of sulfated polysaccharides (named FHC by the authors) from the brown algae *Hormophysa cuneiformis* [74], where a highly sulfated fucogalactan was identified as the main structural motif. An APTT test revealed that the main portion obtained after fractionation by anion-exchange chromatography was only about half as active (2APTT = 7.8 ± 0.3 µg/mL) as enoxaparin (3.8 ± 0.2 µg/mL) and, therefore, had no advantage over many other active fucoidans [74].

Like fucoidans, when SGs are extracted from red algae, they are designated carrageenans. Extracts containing SGs and carrageenans (Figure 7) were isolated from the red algae *Corallina* from the Lebanese coast of Batroun, with total yields of 2.5% and 10%, respectively [101]. APTT assay was used to study the anticoagulant activity of both polysaccharides, the activity of SGs being less potent than that of carrageenans.

Regarding green algae, a large body of literature is available on the isolation and evaluation of the antithrombotic effects of ulvan, a typical sulfated polysaccharide isolated from this kind of algae (Figure 7) [136]. The study of the structural features and anticoagulant activity of the sulfated polysaccharide SPS-CF (ulvan from *Capsosiphon fulvescens*) isolated from green alga *Capsosiphon fulvescens* revealed the presence of xylose and rhamnose as the most prominent monosaccharides, suggesting the presence of glucuronorhamnoxylan, an ulvan [137]. When tested for their anticoagulant activity, APTT and TT were significantly prolonged. In the TT assay, the clotting time of SPS-CF reached up to 31.7 ± 1.3 s (dextran: 14.6 ± 0.2 s and HP: 46.4 ± 2.2 s) and in the APTT assay, SPS-CF also dose-dependently prolonged the clotting time (60.9 ± 1.7 s), although it exhibited weaker activity than HP (76.3 ± 2.0 s (0.3 µg/mL)) but higher in the case of dextran (43.1 ± 1.2 s (200 µg/mL)). Other examples of sulfated polysaccharides containing rhamnose isolated from green algae include the sulfated polysaccharide MSP obtained from *Monostroma angicava* and the respective low-molecular-weight fragments prepared by controlled acid degradation [44]. The polysaccharides were sulfated rhamnans that consisted of →3)-α-l-Rhap-(1→ and →2)-α-l-Rhap-(1→ units with partial sulfation at C-2 of →3)-α-l-Rhap-(1→ and C-3 of →2)-α-l-Rhap-(1→. Regarding their anticoagulant activity, MSP prolonged APTT and TT but failed to prolong PT. APTT and TT prolongation times by MSP were also dose-dependent in vivo. The APTT activity by MSP at 8 mg/kg and 16 mg/kg was considerably stronger than that of HP, but the TT activity was lower. Fibrinolytic activity in vivo was also assayed, and the levels of plasmin degradation products were noticeably enhanced and the level of PAI-1 was effectively reduced by MSP. The activity of the sulfated rhamnan was largely affected by the molecular weight (should be over 12 kDa), and a longer chain was essential to complete thrombin inhibition [44]. Microalgae are also valuable sources of sulfated polysaccharides [119,120,121]. As an example, the sulfated polysaccharides isolated from *Grateloupia livida* (GL) showed significant inhibition of blood coagulation in a dose-dependent manner for both APTT and TT assays [138].

Considering marine invertebrates like echinoderms (sea cucumber and sea urchins), very recent examples also reinforce the importance of specific structural requirement for their controlling functions over anticoagulation factors [47]. Three structurally related sea-urchin (*Strongylocentrotus franciscanus* (*S.f.*), *Lytechinus variegatus* (*L.v.*), and *Echinometra lucunter* (*E.l.*)) derived 3-linked sulfated α-glycans (Figure 8), and their low molecular weight derivatives were screened for their antithrombotic activities [47]. In the APTT assay, *S.f.*, *L.v.*, and *E.l.* were the most active compounds among the six tested, but not as potent as the control unfractionated heparin (UFH). Regarding AT-mediated anti-IIa and anti-Xa, *L.v.* and *E.l.* were the most effective, while *S.f.* was observed to be poorly active. The in vivo venous thrombosis assay using thromboplastin as the thrombogenic stimulus revealed that the six tested compounds presented activity in a dose-response manner, despite their lower efficacy compared to the HP standards. *L.v.*, *S.f.*, and *E.l*. presented the capacity to inhibit 20%, 25%, and 50% of the thrombus weight at the doses of 1.0, 0.5, and 0.25 mg/kg, respectively (UFH and LMWH can prevent 100% of thrombus formation). Concerning platelet aggregation, *E.l.* inhibited approximately 30% of platelet aggregation (against 55% and 30% for UFH and LMWH controls, respectively). This study confirmed the previously reported negative effect of the 2-sulfated fucose and the positive effect of the 2-sulfated galactose on anticoagulation activity in vitro [17,126,139] and demonstrated the importance of this set of structural requirements on antithrombosis in vivo, and further supports the involvement of high-molecular-weight and 4-sulfated fucose in both activities.

An SF was isolated from the sea cucumber *Holothuria albiventer* containing fucose and sulfate in a molar ratio of about 1:0.83 [122]. Structural elucidation revealed that the SF was composed by regular α(1→3) linked hexasaccharide repeating units with a distinctive sulfate substitution pattern (Figure 8). Anticoagulant activity was evaluated by the APTT, PT, and TT assays. Although the concentrations of SFs required to double the APTT (25.79 µg/mL) and TT (115.47 µg/mL) were higher than the positive control LMWH, it showed no significant PT-prolonging activity at the concentrations tested [122]. The intrinsic factor Xase complex (FXase) inhibitory activity revealed a complete inhibition of FXase (half maximal inhibitory concentration, IC_50_ = 71.99 ng/mL) when increasing concentrations of SFs were used, similar to LMWH (IC_50_ = 68.57 ng/mL).

Two other examples of SFs isolated from sea cucumbers (*Holothuria edulis* and *Ludwigothurea grisea*) were recently described. These examples exhibit a novel structural motif for this type of polysaccharides composed of a central core of regular α(1→3) and α(1→2)-linked tetrasaccharide repeating units together with a fucose residue as a side chain (Figure 8), which may contribute to anticoagulant activity [128]. The APTT, PT, and TT were measured and compared with the same activities of UFH and DS. Although both had similar APTT-prolonging activity (aprox. 10 HP U/mg), SFs did not affect PT and TT in human plasma at the concentrations tested (1–100 µg/mL). Antithrombin activity in the presence of HP cofactor II and anti-FXa and antithrombin activity mediated by AT were also examined with chromogenic substrates. Essentially complete inhibition of thrombin activation by HP cofactor II was achieved with increased concentrations of SFs. Further, they displayed significantly weaker anti-FXa and antithrombin activity mediated by AT rather than HP and LMWH (positive controls). Strong inhibition of thrombin (IC_50_ = 0.5–0.7 µg/mL) was shown in the presence of HP cofactor II with DS as a positive reference, while their thrombin and FXa inhibition activities mediated by AT were much weaker. These assays indicate that SFs strongly inhibit blood clotting through the intrinsic pathways of the coagulation cascade, and the mechanism of action was ascribed to the selective inhibition of thrombin activity by HP cofactor II. Interestingly, although also having high molecular weights, these SFs do not induce platelet aggregation, contrary to the SFs from marine algae [128].

### 3.2. Marine Antithrombotics Other than Polysaccharides

Polysaccharides constitute one of the most studied molecules as potential antithrombotic agents, due to their similarity to HP. However, several strategies have been attempted in order to overcome HP’s poor bioavailability [7]. This inconvenience could be surpassed with another type of compounds that could be orally active with or without simple modifications. Several compounds of different chemical natures (peptides, terpenes, alkaloids, polyphenols, steroids, and polyketides), with interesting antithrombotic activities, proved to be active in in vitro and/or in in vivo assays (Table 2). The targets on coagulation cascade (A) and on platelets (B) that were disclosed for some of these marine antithrombotics described in the following subsections are represented in Figure 9.

#### 3.2.1. Peptides

Eight peptides and three proteins were isolated from different marine organisms, such as sponges, starfish, bivalves, fish, and also from a marine echiuroid worm, and were evaluated for their anticoagulant activity and/or for their effects on platelet aggregation (Figure 9A,B).

Plancinin, a peptide with a molecular weight of 7.5 kDa, isolated in 1996 from the starfish *Acanthaster planci*, was incubated with human platelets in the presence of ADP but did not exhibit platelet aggregation activity. However, plancinin showed anticoagulant activity in a dose-dependent manner [140]. Fibrin formation time was prolonged by plancinin (25 μg/mL) and compared with HP (0.08 units). Both caused a clotting time of 150 s (1 unit HP corresponded to 333 μg plancinin). In vivo bleeding time in mice was significantly prolonged by plancinin (at 1–2 h after administration) and by HP (at only 1 h after administration), and 21 units plancinin and 570 units HP were found to be necessary to cause a 200% increase. These results showed that plancinin has the highest anticoagulant activity in vivo [140]. Concerning the anticoagulant tests, plancinin (at 200 µg/mL) prolonged APTT and PT to approximately 110 s and 30 s, respectively (control APTT of 31.2 s and PT of 11.7 s). On the other hand, plancinin did not affect TT. Further studies proved that plancinin could inhibit the prothrombinase complex by action on the activation step of prothrombin and FX [46].

Four linear peptides containing octahydroindole systems (Figure 10), dysinosin A, isolated from a sponge of the family Dysideidae, and dysinosins B-D, obtained from the sponge *Lamellodysidea chlorea*, exhibited activity in thrombin and FVIIa, serine protease enzymes of the blood coagulation cascade (Figure 9A) [38,141]. Dysinosins A–D exhibited inhibitory activity against FVIIa (*k*i values of 0.108 μM, 0.090 μM, 0.124 μM, and 1.320 μM, respectively) and thrombin (*k*i values of 0.452 μM, 0.170 μM, 0.550 μM, and >5.1 μM, respectively) [38]. Comparing with dysinosins A–C, the desulfated dysinosin D showed 10 times less potency against both serine protease enzymes, indicating that the sulfate group contributes to both FVIIa and thrombin binding. The presence of a sugar moiety in dysinosin B (Figure 10) was associated with a slightly increasing inhibition of FVIIa and decreased selectivity for thrombin, compared with dysinosins A and C [38]. In contrast to thrombin, no currently approved anticoagulant agents specifically target FVIIa. Dual inhibition of FVIIa and thrombin could be a promising strategy [11].

In 2008, the *Urechis unicinctus* anticoagulant peptide (UAP) was isolated in from the marine echiuroid worm *U. unicinctus*, with a 3.3 kDa molecular weight [33]. Even though this peptide did not prolong the PT and TT clotting times, the APTT was prolonged in a dose-dependent manner, reaching 192.8 s in the presence of 1.0 mg/mL of UAP (control clotting time of 32.3 s). The FIXa activity in normal plasma decreased by the addition of UAP in a dose-dependent manner (IC_50_ = 42.6 µg/mL). Additional studies concluded that this peptide binds specifically to the FIXa (Figure 9A) [33].

An oligopeptide was isolated in 2009 from the edible parts of the blue mussel *Mytilus edulis*, with approximately 2.5 kDa molecular mass, and the *M. edulis* anticoagulant peptide (MEAP) showed prolongation in the normal clotting time on APTT (to 321 s at 100 μg/mL) and on TT (to 81.3 s at 100 μg/mL), in a dose-dependent manner [35]. In other assays, MEAP inhibited the amidolytic activation of FX, in a dose-dependent manner, with IC_50_ of 13.6 µg/ mL, and also delayed the catalytic conversion of prothrombin to thrombin in the prothrombinase complex with an IC_50_ of 42.9 µg/mL (Figure 9A) [35].

In 2016, a novel anticoagulant peptide was isolated from the seaweed *Porphyra yezoensis* and exhibited a dose-dependent prolongation of APTT from 35 s to 320 s, at a concentration of 3.0 µM and an IC_50_ of 0.3 µM. Nevertheless, this peptide failed to prolong PT and TT [142].

The first anticoagulant protein (26.0 kDa) from marine bivalves was isolated in 2002 from the blood ark shell *Scapharca broughtonii* and, although not exhibiting activity in PT assay, this protein prolonged the clotting time from 32 s to 325 s in the APTT test, at 100 μg/mL, suggesting that it could inhibit a specific factor in the intrinsic pathway. Further tests concluded that this protein inhibited the FIX in the intrinsic pathway of the blood coagulation cascade [41].

A single-chain monomeric protein (12.01 kDa), designated yellowfin sole anticoagulant protein (YAP), was extracted in 2005 from the marine fish, yellowfin sole *Limanda aspera* and exhibited anticoagulant and antiplatelet properties [40]. YAP prolonged APTT in a dose-dependent manner, prolonging the clotting time up to 300 s at 10 µM. However, when tested in PT and TT, neither were affected, suggesting that YAP is a specific inhibitor of the intrinsic pathway of coagulation. Further tests of the inhibitory effect on specific clotting factors in the intrinsic pathway of coagulation showed complete inhibition of FXIIa activity at YAP concentration of 1.5 µM (Figure 9A). Tests on the effect of YAP on platelet aggregation and adhesion were also developed. YAP inhibited, in a dose-dependent manner, platelet aggregation in ADP or thrombin-induced PRP, exhibiting complete inhibition at 660 µM and 600 µM, respectively (Figure 9B) [40]. Simultaneous inhibition of coagulation and platelet aggregation is already known in nature being performed by hematophagous animals, with their saliva containing antithrombotic agents that prevent platelet aggregation [143] and coagulation [144]. Combination of an anticoagulant and an antiplatelet drug proved to be beneficial and is commonly advised in antithrombotic therapy for patients with acute arterial thrombosis [11], with a compound with both activities being a promising antithrombotic alternative [11].

The *Tegillarca granosa* anticoagulant protein (TGAP), isolated in 2007 also from a marine bivalve, granulated ark *T. granosa*, with 7.7 kDa molecular weight, prolonged in a dose-dependent manner the TT clotting time from 11.6 s to 112.8 at 2 mg/mL but not APTT or PT. Further specific factor inhibitory assays concluded that TGAP could inhibit the FVa in the thrombin-formation pathway (essential in converting prothrombin into thrombin) and could inhibit the interaction between prothrombin and FVa, in a dose-dependent manner with an IC_50_ value of 77.9 nM [34].

#### 3.2.2. Terpenes

Eight terpenes with antithrombotic activity (Figure 11) were disclosed by different assays. Halisulfate and suvanine, two sesterterpenes isolated in 1993 from the sponge *Coscinoderma mathewsi* Lendenfield, showed inhibition against the serine proteases thrombin and trypsin [145]. Halisulfate was isolated with the counterion 1-methylherbipoline and exhibited IC_50_ values >100 and 25 μg/mL of inhibition against thrombin and trypsin, respectively [145]. Both salts, 1-methylherbipoline suvanine and sodium suvanine, were isolated and exhibited an inhibitory effect against thrombin (IC_50_ values of 27 and 9 μg/mL, respectively) and against trypsin (IC_50_ values of 12 and 27 μg/mL, respectively) (Figure 9A) [145].

Sargahydroquinoic acid (SHQA) and sargaquinoic acid (SQA) are major constituents of *Sargassum micracanthum* [146] and *Sargassum yezoense* [147] and were first isolated in 2008. SHQA and SQA showed inhibitory collagen-induced platelet aggregation in a dose-dependent manner (Figure 9B) [45]. The percentage of inhibition by SQA was 94.0% at 40 μg/mL, which was much stronger than the effect observed with SHQA (44% at 40 μg/mL) that exhibited a similar effect as the positive control, aspirin (43.0% at 40 μg/mL) [45]. Additionally, in vivo antithrombotic effects of both compounds were evaluated by the recovery time from paralysis. SHQA and SQA showed a fast recovery time from paralysis in the mouse pulmonary thromboembolism model (14.7 min and 6.8 min, respectively), compared to the positive control, aspirin (54.3 min). The authors hypothesized that, with these results, SHQA and SQA are promising lead structures for antiplatelet agents [45].

Dolastane diterpene, isolated in 2011 from the marine brown algae *Canistrocarpus cervicornis* [37] was shown to inhibit ADP- or collagen-induced rabbit PRP aggregation, in a concentration-dependent manner (Figure 9B) [37]. The inhibition was complete at 80 and 100 µM of dolastane diterpene on platelet aggregation induced by ADP and induced by collagen, respectively, with an IC_50_ around 35 µM for both inducers [37]. Concerning the coagulation effects, dolastane also inhibited coagulation induced by thrombin, in a concentration-dependent manner. At a concentration of 90 µM, dolastane inhibited 95% of coagulation in human plasma (IC_50_ of 25 µM) and 64% in bovine fibrinogen (IC_50_ of 45 µM) [37].

Dichotomanol and pachydictyol A/isopachydictyol, obtained in 2014 from the brown alga *Dictyota menstrualis* [148], were investigated for their effects on ADP- or collagen-induced aggregation of PRP and on collagen- or thrombin-induced aggregation of WP. Dichotomanol inhibited, in a concentration-dependent manner, ADP- or collagen-induced platelet aggregation in PRP, with IC_50_ values of 0.31 mM and 1.06 mM, respectively (Figure 9B). However, when tested on WP, the inhibition was not sufficient to obtain an IC_50_ (at 0.32 mM inhibited 15% and 30% aggregation induced by collagen or thrombin, respectively). In complete contrast, pachydictyol A/isopachydictyol inhibited aggregation induced by collagen or thrombin on WP with IC_50_ values of 0.12 mM and 0.25 mM, respectively, while when tested on collagen- and ADP-induced aggregation in PRP inhibition, it was not sufficient to obtain an IC_50_ (at 1.38 mM inhibited 15% and 20% aggregation induced by collagen or ADP, respectively) [148]. Tests with both PRP and WP are important to evaluate if any component of plasma will interfere with the effect of the tested compounds [148]. Concerning coagulation times, APTT, PT, and fibrinogen coagulation were evaluated. Dichotomanol, at a concentration of 1.3 mM, delayed the PT (41.0 s) and APTT (139.9 s) coagulation tests, while pachydictyol A/isopachydictyol A, at 1.4 mM, only delayed the APTT (87.6 s) coagulation test. The diterpenes, dichotomanol, and pachydictyol A/isopachydictyol A inhibited the coagulation of fibrinogen induced by thrombin in 86.4 s and 66.7 s, respectively [148].

#### 3.2.3. Alkaloids

Marine alkaloids have already provided valuable drugs to the market, such as the marine antineoplastic alkaloid, trabectedin (ET-743, Yondelis^®^) [149], and are a fruitful source of bioactive compounds. Concerning antithrombotic activity, zoanthamine-type alkaloids showed promising antiplatelet activities. In vitro biological assays of seven zoanthamine-type alkaloids (Figure 12), isolated in 2003 from marine zoanthids belonging to the *Zoanthus* genus, were performed, focusing on human platelet aggregation induced by several stimulating agents, namely thrombin, collagen, or arachidonic acid (AA) [150].

11-Hydroxyzoanthamine strongly inhibited platelet aggregation induced by thrombin, collagen, and AA, with a percentage of inhibition of approximately 85%, 82%, and 100%, respectively, at 1 mM (Figure 9B). On the other hand, at the same concentration, compounds 3-hydroxynorzoanthamine and 11-hydroxynorzoanthamine (Figure 12) exhibited a more selective effect towards the aggregation induced by collagen (approximately 85% and 100%, respectively) and AA (approximately 82% and 100%, respectively), in contrast to the effects obtained with thrombin-induced aggregation (approximately 25% and 50%, respectively). 30-Hydroxynorzoanthamine also showed a more selective effect towards aggregation induced by collagen (60% inhibition at 1 mM) and AA (35% inhibition at 1 mM) but overall behaved as a less powerful antiplatelet agent. Another series of zoanthamines behaved as selective inhibitors of collagen-induced platelet aggregation, namely oxyzoanthamine (approximately 50% inhibition at 0.5 mM), zoanthenol (approximately 70% inhibition at 0.5 mM), and epioxyzoanthamine (approximately 14% inhibition at 0.5 mM), which cause little or no effect against AA- or thrombin-induced aggregation (Figure 9B) [150]. This pharmacological study demonstrates that small structural variations have a marked influence on the antithrombotic activity of zoanthamines [150].

#### 3.2.4. Polyphenols

Five polyphenols isolated from different marine organisms were evaluated through different assays (Figure 13).

Eckol and dieckol, two purified phlorotannins, isolated from *Eclonia kurome* in 1985, exhibited fibrinolytic activity inhibiting the main plasmin inhibitors, α_2_-macroglubin (IC_50_ of 2.5 and 5.0 µg/mL, respectively) and α_2_-antiplasmin (IC_50_ of 1.6 and 0.8 µg/mL, respectively) [42]. In 2012, further tests were assayed, and eckol (10 μM) and dieckol (10 μM) [36] prolonged coagulation times in APTT (71.6 s and 82.5 s, respectively) and in PT (26.9 s and 29.4 s, respectively) tests weaker than HP (at 1.5 μg/mL, the coagulation time was >300 s in the APTT and 62.4 s in the PT assay). These effects suggested that eckol and dieckol are inhibitors of the common pathway and of the extrinsic pathway of coagulation (Figure 9A) [36]. The inhibitory effect on thrombin and FXa activities was also measured, in order to elucidate their anticoagulant mechanism action, in the absence or presence of ATIII. The two phlorotannins only inhibited the amidolytic activity of thrombin in the absence of ATIII (Figure 9A) [36]. The author justified the small differences between eckol and dieckol anticoagulant activity as being attributed to the different number and position of hydrogen donating hydroxyl groups in the molecules of these compounds [36].

Phlorofucofuroeckol A, a phlorotannin isolated in 1990 from the brown alga *Ecklonia kurome*, showed fibrinolytic activity in plasma by inhibition of the main plasmin inhibitors, α_2_-macroglubin and α_2_-antiplasmin, with IC_50_ values of 1.0 μg/mL and 0.3 μg/mL, respectively [43].

Aplysillin A, a disulfated and dibrominated 1,4-diphenyl-1,3-butadiene, was isolated in 1995 from the sponge *Aplysina fistularis fulva* (Pallas) and was able to inhibit the binding of thrombin to platelet membranes with an IC_50_ value of 20 μM (Figure 9B) [151].

Phloroglucinol, a common monomer unit of the polymer phlorotannin, was found in *Ecklonia* species in 1997 [152]. Phloroglucinol (10 μM) prolonged coagulation times to 93.1 s and 34.1 s in the APTT and PT tests, respectively, comparing with the control (35.2 s on APTT and 17.2 s on PT). Although weaker than HP (at 1.5 μg/mL APTT >300 s and PT 62.4 s), the prolongation of APTT and prolongation of PT by phloroglucinol suggested the inhibition of the common pathway and inhibition of extrinsic pathway of coagulation, respectively [39]. The inhibitory effect on thrombin and FXa activities was measured to elucidate the anticoagulant mechanism of the phloroglucinol, in the absence or presence of ATIII. Phloroglucinol inhibited the amidolytic activity of thrombin, in the absence of ATIII, in a dose-dependent manner. However, in the presence of ATIII, no inhibition was observed [39], leading to the hypothesis that phloroglucinol anticoagulant activity was due to direct inhibition of thrombin activity. Phloroglucinol also showed a direct inhibitory effect on FXa activity, in the absence of ATIII, and no effect was observed in the presence of ATIII (Figure 9A) [39]. In 2012, further studies concerning antiplatelet aggregation showed that phloroglucinol (2.5–25 μM) inhibited AA-induced platelet aggregation in a concentration dependent manner. Phloroglucinol (10, 25, and 50 μM) also inhibited cyclooxygenase (COX) 1 enzyme activity by 45%, 64%, and 74%, respectively, and COX-2 by 49%, 53%, and 72%, respectively. In vivo bleeding time in mice was significantly prolonged by phloroglucinol (maximal bleeding time approximately 180 s, at 0.5 μM/ mice), compared to the control mice that had an average tail bleeding time of 84 s (Figure 9A,B) [153].

A combination of thrombin and FXa inhibitory activities in a single, synthetic, orally active compound with low molecular weight is a promising approach to antithrombotic therapy [11]. In fact, a study found synergistic effects in the simultaneous inhibition of thrombin and FXa, in a rabbit arteriovenous shunt model of thrombosis [154]. Dual-acting thrombin/FXa inhibitors were already developed by Boehringer Ingelheim [155] and also by Merck & Co pharmaceutical companies [156].

#### 3.2.5. Steroids

Four steroids isolated in 1996 from sponges were investigated for their antiplatelet actions (Figure 14). Contignasterol, isolated from the marine sponge *Petrosia contignata*, exhibited inhibitory activity against platelet aggregation induced by PAF and also induced by collagen. The inhibition of platelet aggregation was found to be complete when induced by PAF (at 50 µg/mL) and collagen (at 20 µg/mL) (Figure 9B) [157].

Eryloside F, a disaccharide of the steroidal carboxylic acid penasterol obtained in 2000 from marine sponge *Erylus formosus*, exhibited functional activity in an in vitro human WP aggregation assay [158]. Eryloside F was able to inhibit the platelet aggregation induced by a thrombin receptor activating peptide, SFLLRN, and a stable thromboxane A2 mimetic, U-46619 with IC_50_ values of 0.3 and 1.7 mg/mL, respectively. Eryloside F was also tested against platelet aggregation induced by thrombin, but potency was to low relative to its potency against SFLLRN-induced platelet aggregation [158].

In 2003, two sulfated sterols were isolated from the marine sponge *Topsentia sp.* (Halichondriidae), the halistanol trisulfate and the Sch-572423, which were identified as antiplatelet agents acting through P2Y_12_ receptors (Figure 9B) with IC_50_ values of 0.48 and 2.2 µM, respectively [159]. P2Y_12_ is one of the most important ADP receptors and is the target of ticlopidine, clopidogrel, prasugrel, ticagrelor, and cangrelor [160].

#### 3.2.6. Polyketides

Three polyketides were isolated from different species of marine sponges and tested for their antiplatelet activity (Figure 15).

Okadaic acid, a polyether derivative of a C_38_-fatty acid, originally isolated from the marine sponges of the genus *Halichondria okadai* in 1981 [161], is a marine sponge toxin that inhibits thrombin-induced aggregation of rabbit platelets [162] and was identified as a potent inhibitor of protein phosphatases type 1 and type 2A [163]. Okadaic acid inhibited thrombin-induced aggregation, ATP release, and the increase in cellular Ca^2+^ induced by thrombin in 8.1%, 17.2%, and 20.7%, respectively, at a concentration of 1 μM [162].

Xestospongin/araguspongine, isolated in 2003 from the marine sponge *Xestospongia* sp., inhibited collagen-induced aggregation and epinephrine-induced aggregation by 91.6% at 200 μg/mL and by 97.4% at 200 μg/mL, respectively, while 5,6-dibromotryptamine, from the sponge *Aplysina* sp., inhibited collagen- and epinephrine-induced aggregation by 77.8% at 200 μg/mL and by 92.0% at 200 μg/mL [164].

## 4. Conclusions

Cardiovascular diseases are a leading contributor to morbidity and mortality in the 21st century. Demand for new antithrombotic compounds, especially with multitarget actions, is increasing. Oceans constitute a vast source of structurally new and unique biologically active molecules, being a rich source of diverse organisms, containing thousands of described species and many others yet to be discovered. The lower contamination levels of viruses and/or prions in marine organisms in comparison with the risks associated with mammalian HP derivatives is a strength of this source of potential antithrombotic molecules.

Polysaccharides constitute one of the most studied molecules as potential antithrombotic agents; however, oral bioavailability could be an issue, as it happens with HP. Exploring other chemical classes with smaller and/ or different structures could be an advantage in discovering compounds that are potentially orally active.

Herein, 38 non-polysaccharide compounds were presented, from six different chemical classes, such as alkaloids, peptides, polyketides, polyphenols, steroids, and terpenes, with very unique structures and high degrees of molecular substitution, which were studied for their anticoagulant and/or antiplatelet activities in different assays. The majority of these marine antithrombotic compounds were isolated from sponges or algae, with the exception of the antithrombotic peptides, which were extracted from a large diversity of marine species, such as sponges, starfish, bivalves, fish, and marine worms. The antithrombotic activity of more than half of these molecules was described after the year 2000, in which half was described in the last decade. Many marine compounds were studied through both in vitro and in vivo assays.

Several compounds exhibited dual activity, which is an attractive approach in complex etiologies, such as cardiovascular diseases. The combination of an anticoagulant with an antiplatelet drug has already proven its benefits on antithrombotic therapy for patients with acute arterial thrombosis. Three terpenes (dichotomanol, dolastane diterpene, and pachydictyol), one protein (YAP), and one polyphenol (phloroglucinol) exhibited dual anticoagulant and antiplatelet activities. This profile was also found in some polysaccharides (FCS, MM, and *E.l.*). Additionally, dual anticoagulant activity through direct inhibition of serine proteases was exhibited by peptides dysinosins A–C (FVIIa and thrombin) as also by the polyphenols phloroglucinol, eckol, and dieckol (FXa and thrombin). These three serine proteases are crucial coagulation factors and have been pointed out as preferred targets for the development of new antithrombotic drugs.

This review highlights the new scaffolds being disclosed as promising antithrombotic agents and the opportunity of testing novel classes beyond the polysaccharide chemical space. Further studies should be conducted in order to optimize these marine compounds by structural modifications, as well as to better understand their structure–antithrombotic activity relationship in order to develop new solutions for cardiovascular diseases.

## 5. Future of Marine Antithrombotic Molecules

The methods to determine antithrombotic activity are evolving along with the advances of clinical and pharmaceutical analysis. Nevertheless, the majority of the reports presented herein limited their studies to classical clotting assays and conventional coagulation targets. The intervention of the genomic/proteomic era in the investigation of antithrombotics from the sea is still to come. An increase in the studies of molecules from the sea was noted in the last decade, particularly for SGs and fucoidans polysaccharides and for novel frameworks. It is worth noting that MSP from the edible green algae is forthcoming as drug or as a food supplement for human health. Continuous efforts in this field will lead to developments of alga-derived agents with potential clinical uses in thrombosis. So far, the investigation on anticoagulant sulfated rhamnans is still less active than that of galactans and fucoidans, but if more efforts are made, more anticoagulant properties on the sulfated rhamnans should be obtained. Recent studies on polysaccharides highlight the importance of the molecular weight on their antithrombotic activities. Curiously, when observing the most active compounds from other frameworks, most are out of the space of the Lipinski rules and can be a limitation when aiming for oral bioavailability. Nonetheless, knowledge of peptidomimetic chemistry has increased in the last years, and tuning a peptide into a small orally-active molecule is a feasible project. The strength of these novel antithrombotic scaffolds is the high selectivity found in some representatives like the anticoagulant peptides UAP, MEAP, and the protein from the blood ark shell. Therefore, our bet for the future lies in these novel structures that can be the target of medicinal chemists to achieve the desirable bioavailability. One cannot discharge the multitarget compounds; this would be also the case for HP, which already rendered derivatives with different applications. In these complex etiologies, such as cardiovascular diseases, multitarget molecules have proven benefits. Most importantly, one needs to rule out or prove whether polyphenols, like the multitarget phloroglucinol, are not acting as PAINs (pan-assay interference compounds) through an unspecific mechanism of action functioning as reactive chemicals rather than discriminating drugs.

## Figures and Tables

**Figure 1 marinedrugs-17-00170-f001:**
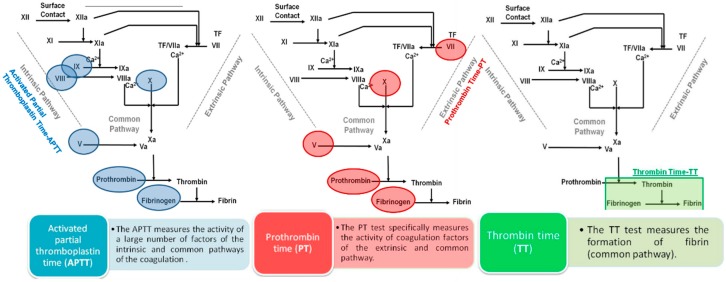
Activated partial thromboplastin time (APTT), prothrombin time (PT), and thrombin time (TT) assays and their link with the classical coagulation cascade. TF—tissue factor.

**Figure 2 marinedrugs-17-00170-f002:**
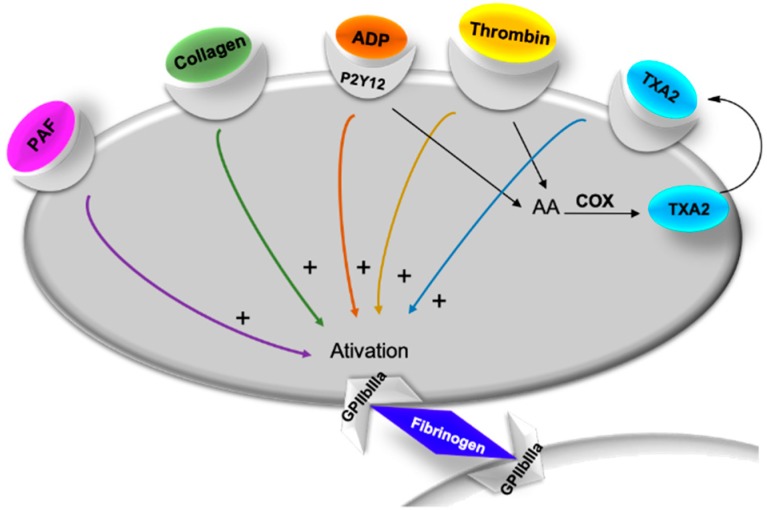
Platelet activation agonists. ADP—adenosine diphosphate; AA—arachidonic acid; ADP—adenosine diphosphate; COX—cyclooxygenase; PAF—platelet activating factor; TXA_2_—thromboxane A2.

**Figure 3 marinedrugs-17-00170-f003:**
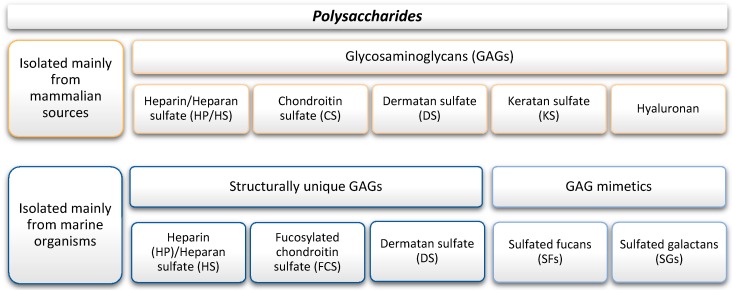
Most representative classes of polysaccharides isolated mainly from mammalian sources and marine organisms.

**Figure 4 marinedrugs-17-00170-f004:**
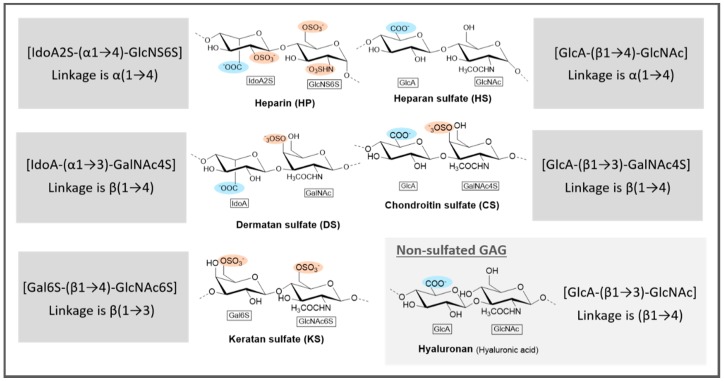
General representation of glycosaminoglycans of mammals. IdoA2S—2-sulfated iduronic acid; GlcNS6S—*N*,6-disulfated glucosamine; GlcA—glucuronic acid; GlcNAc—*N*-acetylglucosamine; GalNAc4S—4-sulfated *N*-acetylgalactosamine; IdoA—iduronic acid; Gal6S—6-sulfated galactose; GlcNAc6S—6-sulfated *N*-acetylglucosamine. Adapted from Reference [50].

**Figure 5 marinedrugs-17-00170-f005:**
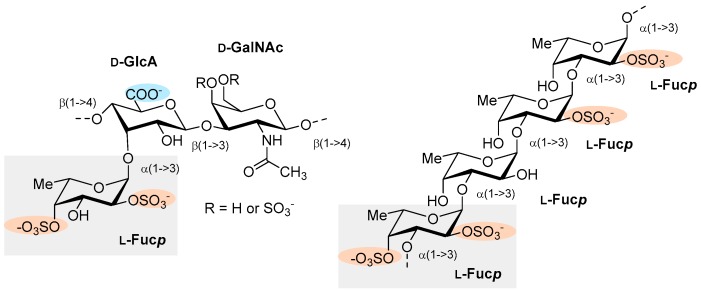
General representation of the anticoagulant molecules from the sea urchin holothurian species *Ludwigothurea grisea*. Adapted from References [17,51].

**Figure 6 marinedrugs-17-00170-f006:**
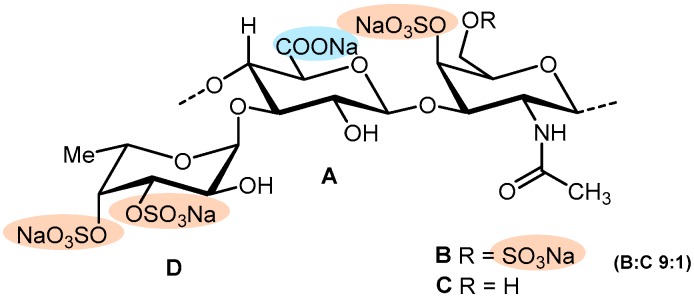
Structure of the repeating trisaccharide units of a fucosylated chondroitin sulfate isolated from sea cucumber *Massinium magnum* (MM) [67].

**Figure 7 marinedrugs-17-00170-f007:**
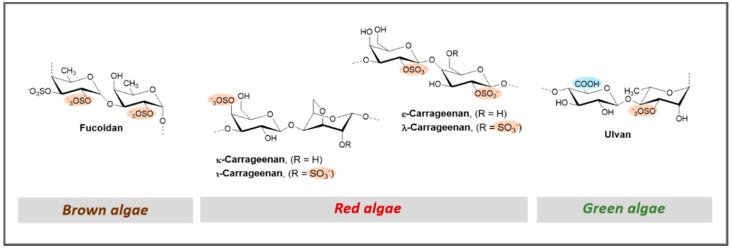
General representation of the structural features of marine sulfated glycosaminoglycans (GAG) mimetics isolated from brown, red, and green algae.

**Figure 8 marinedrugs-17-00170-f008:**
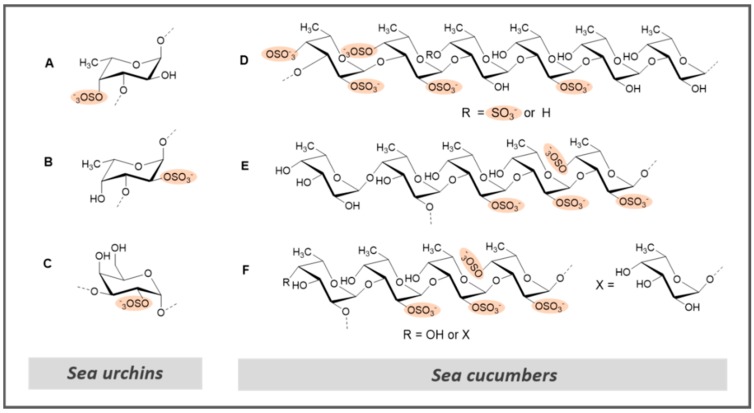
Repeating units of sulfated fucans and sulfated galactans isolated from marine invertebrates proposed by Wu et al. [128]: sea urchin (**A**) 3-linked 4-sulfated α-fucan from the *Lytechinus variegatus*; (**B**) 3-linked 2-sulfated α-fucan from *Strongylocentrotus franciscanus*; (**C**) 3-linked 2-sulfated α-galactan from *Echinometra lucunter* [47]; sea cucumber (**D**) *Holothuria albiventer* [122]; (**E**) *Holothuria edulis*; (**F**) *L. grisea*.

**Figure 9 marinedrugs-17-00170-f009:**
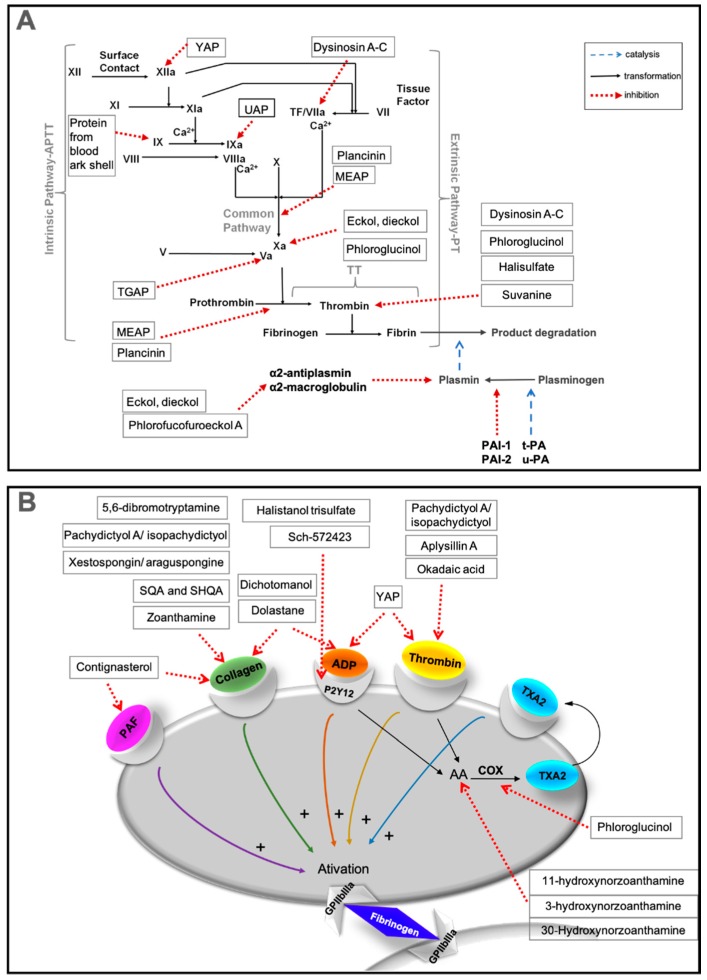
Marine antithrombotics other than polysaccharides and their effects on (**A**) the coagulation cascade; (**B**) the platelet activation. AA—arachidonic acid; ADP—adenosine diphosphate; COX—cyclooxygenase; PAF—platelet activating factor; PAI-1 or -2—plasminogen activator inhibitor 1 or 2; TXA_2_—thromboxane A2; u-/t-PA—tissue-type plasminogen activator.

**Figure 10 marinedrugs-17-00170-f010:**
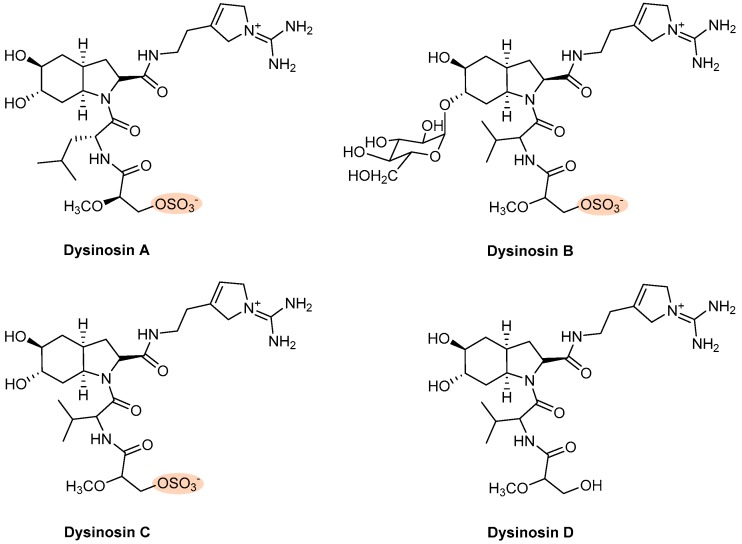
Chemical structures of dysinosin A–D with antithrombotic activity.

**Figure 11 marinedrugs-17-00170-f011:**
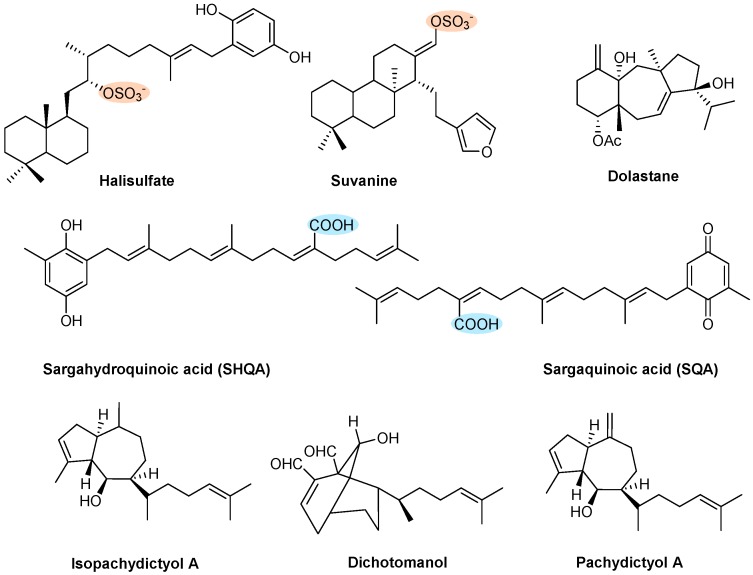
Chemical structures of terpenes with antithrombotic activity.

**Figure 12 marinedrugs-17-00170-f012:**
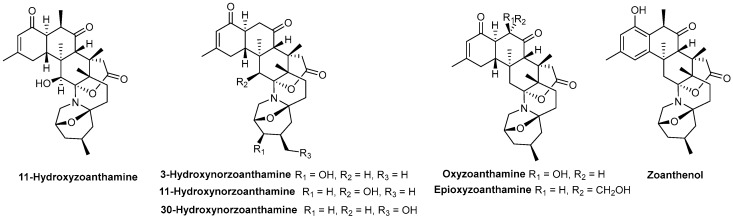
Chemical structures of zoanthamine-type alkaloids with antithrombotic activity.

**Figure 13 marinedrugs-17-00170-f013:**
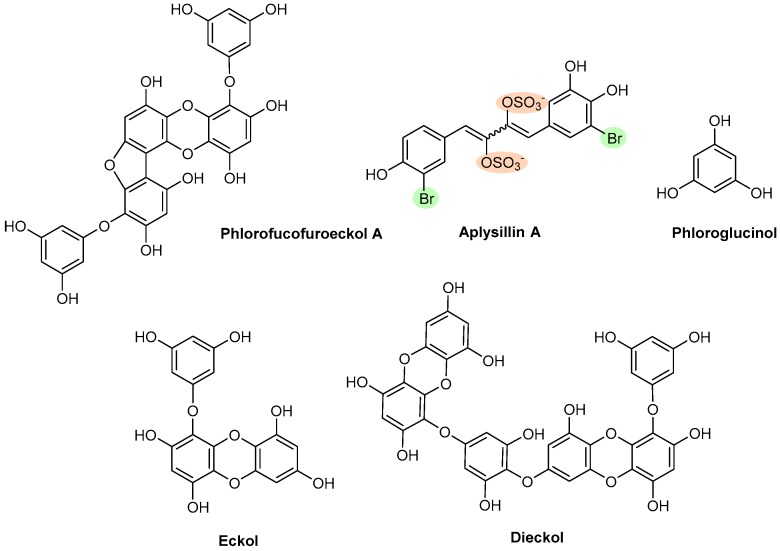
Chemical structures of polyphenols with antithrombotic activity.

**Figure 14 marinedrugs-17-00170-f014:**
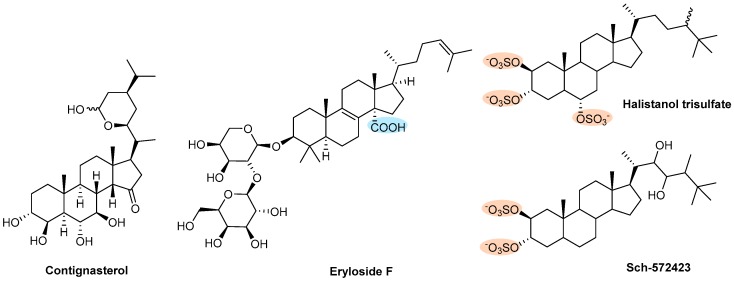
Chemical structures of steroids with antithrombotic activity.

**Figure 15 marinedrugs-17-00170-f015:**
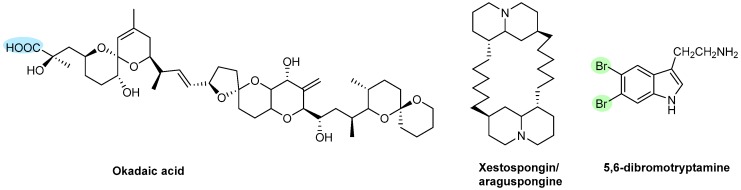
Chemical structures of polyketides with antithrombotic activity.

**Table 1 marinedrugs-17-00170-t001:** Anticoagulant and antiplatelet activities of GAGs and GAG mimetics isolated from marine sources in the last five years (2014–2018). ↑—prolonged/delayed; PS—polysaccharides; ↓—Reduction; Θ—inhibitor; NR—not represented; NT—not tested; NA—not active.

PS Type	Molecular Type	Name(s)	Source	Structure	Antithrombotic Assays		Ref.
					Anticoagulant	Antiplatelet	
Marine GAGs	**Fucosylated chondroitin sulfates**	FCSc FCSt	Sea cucumber (*Cucumaria frondosa* and *Thelenota ananas*)	NR	↑ APTT Θ Thrombin and anti-FXa (in the presence of HP cofactor II) ↑ Bleeding time (in vivo)	NT	[65]
		FCS	Sea cucumber (*Cucumaria japonica*)	NR	↑ APTT Θ Thrombin (in the presence of ATIII)	Θ Collagen and ristocetin-induced platelets aggregation	[66]
		MM	Sea cucumber (*Massinium magnum*)	Figure 6	↑ APTT, and TT Θ Thrombin (in the presence of ATIII)	NA—Platelets aggregation in PRP	[67]
		FCShm	Sea cucumber (*Holothuria Mexicana)*	NR	↑ APTT, and TT	NT	[68]
		HsG	Sea cucumber (*Holothuria scabra*)	NR	↑ APTT, and TT Θ Thrombin and FXa (in the presence of ATIII)	NT	[69]
Marine GAG mimetics	**Sulfated fucans**	FSA (fucoidan)	Brown algae (*Sargassum aquifolium*)	NR	↑ APTT	NT	[135]
		FHC (fucogalactan)	Brown algae (*Hormophysa cuneiformis*)	NR	↑ APTT	NT	[74]
		*L.v.*	Sea urchin (*Lytechinus variegatus*)	Figure 8A	↑ APTT ↑ FXa and thrombin (in the presence of ATIII) Θ Thrombus formation Θ Thrombus formation (in vivo)	NA	[47]
		*S.f.*	Sea urchin (*Strongylocentrotus franciscanus*)	Figure 8B	↑ APTT Θ Thrombus formation Θ Thrombus formation (in vivo)	NA	[47]
		FS	Sea cucumber (*Holothuria albiventer*)	Figure 8D	↑ APTT and TT Θ FXase inhibition	NT	[122]
		*H. edulis*	Sea cucumber (*Holothuria edulis)*	Figure 8E	↑ APTT Θ Thrombin (in the presence of HP cofactor II)	NT	[128]
		*L. grisea*	Sea cucumber (*Ludwigothurea grisea*)	Figure 8F	↑ APTT Θ Thrombin (in the presence of HP cofactor II)	NT	[128]
	**Sulfated galactans**	Carageenan	Red algae (*Corallina*)	NR	↑ APTT	NT	[101]
		*E.l.*	Sea urchin (*Echinometra lucunter*)	Figure 8C	↑ APTT ↑ anti FXa and thrombin (in the presence of ATIII) Θ thrombus formation Θ Thrombus formation (in vivo)	Θ platelet aggregation	[47]
	**Other sulfated PS**	SPS-CF (ulvan)	Green algae (*Capsosiphon fulvescens*)	NR	↑ APTT and TT	NT	[137]
		MSP	Green algae (*Monostroma angicava*)	NR	↑ APTT and TT ↑ APTT and TT (in vivo) ↑ Plasmin degradation products (in vivo) ↓ Level of PAI-1 (in vivo)	NT	[44]
		GP	Microalgae (*Grateloupia livida*)	NR	↑ APTT and TT	NT	[138]

**Table 2 marinedrugs-17-00170-t002:** Antithrombotic activities of peptides, terpenes, alkaloids, polyphenols, steroids, and polyketides isolated from marine sources.

Chemical Class	Name	Source	Structure	Antithrombotic Assays		Ref.
				Anticoagulant	Antiplatelet	
Peptides	**Peptide from seaweed *Porphyra yezoensis***	Seaweed (*Porphyra yezoensis*)	NR	↑ APTT	NT	[142]
	**Plancinin**	Starfish (*Acanthaster planci*)	NR	↑ Fibrin test ↑ APTT and PT Θ Prothrombinase complex ↑ Bleeding time (in vivo)	NA	[140]
	**Dysinosin A–D**	Sponge (*Dysideidae*)	Figure 10	Θ FVIIa, thrombin	NT	[38,141]
	**UAP**	Echiuroid worm (*U. unicinctus*)	NR	↑ APTT Binds to FIXa	NT	[33]
	**MEAP**	Blue mussel (*Mytilus edulis*)	NR	↑ APTT and TT Θ Amidolytic activation of FX ↑ Catalytic conversion of prothrombin to thrombin	NT	[35]
	**Protein from blood ark shell**	Blood ark shell (*Scapharca broughtonii*)	NR	↑ APTT Θ FIX	NT	[41]
	**YAP**	Yellowfin sole (*Limanda aspera*)	NR	↑ APTT Θ FXIIa	Θ Thrombin- or ADP-induced aggregation	[40]
	**TGAP**	Granulated ark (*T. granosa*)	NR	↑ TT Θ FVa Θ Interaction between prothrombin and FVa	NT	[165]
Terpenes	**Halisulfate and Suvanine**	Sponge (*Coscinoderma mathewsi*)	Figure 11	Θ Thrombin and trypsin	NT	[145]
	**SHQA and SQA**	Algae (*Sargassum micranthum* and *Sargassum yezoense*)		Fast recovery time from paralysis (in vivo)	Θ Collagen-induced aggregation	[146,147]
	**Dolastane**	Brown algae (*Canistrocarpus cervicornis*)		Θ Thrombin-induced coagulation	Θ ADP- or collagen-induced aggregation	[37]
	**Dichotomanol**	Brown algae (*Dictyota menstrualis*)		↑ APTT and PT Θ Thrombin-induced fibrinogen	Θ ADP- or collagen-induced aggregation	[148]
	**Pachydictyol A/Isopachydictyol**	Brown algae (*Dictyota menstrualis*)		↑ APTT Θ Thrombin-induced fibrinogen	Θ ADP- or thrombin-induced aggregation	[148]
Alkaloids	**11-Hydroxyzoanthamine**	Zoanthids (*Zoanthus Z. nymphaeus and an unidentified species of Zoanthus* sp.)	Figure 10	NT	Θ Thrombin-, collagen- or AA-induced aggregation	[150]
	**3 and 11-Hydroxynorzoanthamine**			NT	Θ Collagen- or AA-induced aggregation	[150]
	**Oxyzoanthamine, Zoethenol and Epioxyzoanthamine**			NT	Θ Collagen-induced aggregation	[150]
Polyphenols	**Phlorofucofuroeckol A**	Brown algae *(Ecklonia Kurome)*	Figure 13	Θ Plasmin inhibitors	NT	[43]
	**Aplysillin A**	Sponge (*Aplysina fistularis fulva*)		Θ Thrombin to platelet membranes	NT	[151]
	**Phloroglucinol**	Brown algae (*Ecklonia species*)		↑ APTT and PT Θ Amidolytic activity of thrombin, in the absence of ATIII Θ FXa in the absence of ATIII ↑ Bleeding time (in vivo)	Θ AA-induced aggregation Θ COX-1 enzyme	[152,153]
	**Eckol and Dieckol**	Brown algae (*Ecklonia kurome*)		↑ APTT and PT Θ Amidolytic activity of thrombin, in the absence of ATIII Θ FXa in the absence of ATIII Θ Plasmin inhibitors	NT	[36]
Steroids	**Contignasterol**	Sponge (*Petrosia contignata*)	Figure 14	NT	Θ PAF- or collagen-induced aggregation	[156]
	**Eryloside F**	Sponge (*Erylus formosus*)		NT	Θ SFLLRN and U-46619 induced aggregation	[158]
	**Halistanol trisulfate and Sch-572423**	Sponge (*Topsentia* sp.)		NT	Θ Platelet aggregation acting through P2Y_12_ receptors	[159]
Polyketides	**Okadaic acid**	Sponges (*Halichondria okadai*)	Figure 15	NT	Θ Thrombin-induced aggregation Θ Protein phosphatases type 1 and type 2A Θ ATP release and increased cellular Ca^2+^	[162]
	**Xestospongin/Araguspongine and 5,6-Dibromotryptamine**	Sponge (*Xestospongia sp.*)		NT	Θ Collagen- or epinephrine-induced aggregation	[164]

↑—prolonged/delayed Θ—inhibitor; SFLLRN—thrombin receptor activating peptide; U-46619—stable thromboxane A2 mimetic; NR—not represented; NT—not tested; NA—not active.

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
