# Peer review of "Antithrombotics from the Sea: Polysaccharides and Beyond"

_marinedrugs, 2019, doi:10.3390/md17030170_

Round 1
Reviewer 1 Report
Quite generally, the authors wrote a very interesting review highlighting the potential of marine organisms in terms of antithrombotic effet and beyond. The paper is rich, clear and well illustrated.
However, to complete the paper, it should be relevant to update the recent progress observed this last decade on microalga emerging as great source for the synthesis of sulfated polysaccharides and other bioactive molecules as well. The addition of some information concerning the GAGs derived microalga, for example, should enrich this good work.
- A liste of abbreviations is also required to better understand the review.
Author Response
REVIEWER 1
Comments and Suggestions for Authors
Quite generally, the authors wrote a very interesting review highlighting the potential of marine organisms in terms of antithrombotic effect and beyond. The paper is rich, clear and well illustrated.
However, to complete the paper, it should be relevant to update the recent progress observed this last decade on microalga emerging as great source for the synthesis of sulfated polysaccharides and other bioactive molecules as well. The addition of some information concerning the GAGs derived microalga, for example, should enrich this good work.
Reply: We thank the reviewer for this suggestion and the following paragraph and references of three review articles* were added to the text in line 321 …” Microalgae are also valuable sources of sulfated polysaccharides [119-121]. As an example, the sulfated polysaccharides isolated from Grateloupia livida (GL) showed significant inhibition of blood coagulation in a dose-dependent manner for both APTT and TT assays [138].
Since the literature examples, to the best of our knowledge, were reported before 2014 and, therefore, are outside the period (2014-2018) that we considered for the marine GAGs and marine GAG mimetics, they were not discussed in detail. Also, the article deals only with the isolation and antithrombotic activities of marine polysaccharides and other molecules and does not approach the synthesis of sulfated polysaccharides.
*Review articles added to the manuscript:
119. Raposo, M.F.d.J.; De Morais, R.M.S.C.; Bernardo de Morais, A.M.M. Bioactivity and applications of sulphated polysaccharides from marine microalgae. Mar. Drugs 2013, 11, 233.
120. Raposo, M.F.d.J.; de Morais, A.M.M.B. Microalgae for the prevention of cardiovascular disease and stroke. Life Sciences 2015, 125, 32-41. (10.1016/j.lfs.2014.09.018)
121. Yu, Y.; Shen, M.; Song, Q.; Xie, J. Biological activities and pharmaceutical applications of polysaccharide from natural resources: A review. Carbohydr. Polym. 2018, 183, 91-101. (10.1016/j.carbpol.2017.12.009)
- A list of abbreviations is also required to better understand the review.
Reply: We agree with the reviewer. A list of abbreviations was added at the end of the manuscript.
Reviewer 2 Report
I think the article "Antithrombotics from the sea: polysaccharides and beyond" is well written. The authors need to expand their discussion in detail and incorporate newly published references.
Author Response
I think the article "Antithrombotics from the sea: polysaccharides and beyond" is well written. The authors need to expand their discussion in detail and incorporate newly published references.
Reply: We thank reviewer #2 for his/her valuable comment. The discussion on the Marine GAGs and Marine GAG mimetics was detailed to be consistent with the remaining sub-sections that were already quite detailed. All the changes are highlighted in the text. Polysaccharides sub-section already has the latest published references since it covers the last five years.
Reviewer 3 Report
The manuscript is not suitable for scientific assessment due to the multiple English missuses/mistakes. I can revise once this aspect has been taken care of.
Author Response
The manuscript is not suitable for scientific assessment due to the multiple English missuses/mistakes. I can revise once this aspect has been taken care of.
Reply: As suggested by reviewer #3, the manuscript was revised by MDPI English editing servisse.
Reviewer 4 Report
Revision of manuscript marinedrugs-448201 entitled “Antithrombotics from the sea: polysaccharides and beyond”.
Dear authors,
The topic of the present manuscript has scientific relevance. However, in my opinion the current manuscript has several issues for the reader of Marine drugs. Please find below few comments about the current manuscript hoping to improve it.
Kind regards
Comments
1. Aim of the review paper
In my opinion the aim of this review is a bit narrow “isolation and activity of antithrombotic small molecules from marine organisms is missing” (line 50), that is not fully explained in the conclusions and does not cover many interesting points of these molecules, decreasing the impact of the work.
2. Structure of the manuscript
The manuscript and understanding of the content for the readers will improve by including multiple sections. I.e. “Measurement of antithrombotic activities“, “Sources and chemical structure of main compounds”, “”novel polysaccharides”, “novel peptides”…”Future of marine antithrombotic molecules”.
3. Tables
Tables will be useful to summarize key features for the readers (i.e. source, IC50, activities…) that will be fully detailed in the main manuscript.
3. Abbreviation list.
The authors used a huge amount of abbreviations in the manuscript, being some of them not mentioned in the main manuscript I.e. LMWH was mentioned for the first time in line 128 and not explained before. Moreover, due to the huge amount of abbreviations and names of fractions of molecules that previous authors used in the original publications, it is really difficult to read the main message of the manuscript.
5. Figures not explained.
“The targets on coagulation cascade (A) and on platelets (B) that were disclosed for some of these marine antithrombotics are represented at the end of this sub-chapter in Chart 1 for the reader convenience” (lines 245-247). This is the only explanation that Chart 1 that is one full page with 2 figures is given to the readers. As I previously mentioned, this information could be placed in a section on its own and fully explained as Marine drugs has no limit of words to submit manuscripts.
6. General preparation and English editing
The manuscript will require English editing. Few examples are listed below, although this is not an exhaustive list.
Line 15: “some advantages as a renew source of potential”
Lines 44-47: “Although over the last decades polysaccharides have been identified as the most therapeutically explored metabolites and suppliers of new antithrombotic agents, in fact, the less explored small molecules have proven to be also an excellent starting point for the development of new and orally effective drugs.”
Lines 50-53: “Herein, we perform a run-up on some major representatives and newest examples of the bioactive sulfated polysaccharides [covering the last five years (2014-2018)] and to include other molecules particularly with potential oral bioavailability.”
Author Response
Revision of manuscript marinedrugs-448201 entitled “Antithrombotics from the sea: polysaccharides and beyond”.
Dear authors,
The topic of the present manuscript has scientific relevance. However, in my opinion the current manuscript has several issues for the reader of Marine drugs. Please find below few comments about the current manuscript hoping to improve it.
Kind regards
Reply: We thank the reviewer #4 for all the valuable comments. We have revised the manuscript accordingly and the changes are highlighted in the text and explained point by point below.
Comments
1. Aim of the review paper
In my opinion the aim of this review is a bit narrow “isolation and activity of antithrombotic small molecules from marine organisms is missing” (line 50), that is not fully explained in the conclusions and does not cover many interesting points of these molecules, decreasing the impact of the work.
Reply: We acknowledge reviewer # 4 comment. The sentence was not completely clarifying and it was changed to …”While comprehensive and updated reviews can be found in the literature regarding marine sulfated polysaccharides with antithrombotic activities [9-31], reports of marine antithrombotic small molecules are still disperse sparse in the literature. Therefore, we performed a run-up review of some of the major representatives and newest examples of bioactive polysaccharides reported in the last five years, and a systematic compilation of small molecules isolated from marine organisms with antithrombotic activities.“… (Line 49)
2. Structure of the manuscript
The manuscript and understanding of the content for the readers will improve by including multiple sections. I.e. “Measurement of antithrombotic activities“, “Sources and chemical structure of main compounds”, “”novel polysaccharides”, “novel peptides”…”Future of marine antithrombotic molecules”.
Reply: We acknowledge and agree with the reviewer # 4’s concern. A new section (Section 2 - Methods used to evaluate antithrombotic activities of marine compounds) was added to the manuscript where we detail all the methods that have been used to evaluate antithrombotic activities of the Marine Compounds with Antithrombotic Activity cited in the article. Two tables (Table 1 and 2) were added in order to summarize the sources, chemical structure, and activities of all the cited compounds. Also, a new section was added at the end of the manuscript (Section 5 - Future of marine antithrombotic molecules) where we envision the future of some of the derivatives highlighting novel structures and multitarget compounds. We believe these changes helped in clarifying several aspects and improved the quality of the work.
3. Tables
Tables will be useful to summarize key features for the readers (i.e. source, IC50, activities…) that will be fully detailed in the main manuscript.
Reply: Two tables (Table 1 and 2) were added in order to summarize the sources, chemical structure, and activities of all the cited compounds.
3. Abbreviation list.
The authors used a huge amount of abbreviations in the manuscript, being some of them not mentioned in the main manuscript I.e. LMWH was mentioned for the first time in line 128 and not explained before. Moreover, due to the huge amount of abbreviations and names of fractions of molecules that previous authors used in the original publications, it is really difficult to read the main message of the manuscript.
Reply: We agree with the reviewer. A list of abbreviations was added at the end of the manuscript, including names of fractions of molecules that previous authors used in the original publications.
5. Figures not explained.
“The targets on coagulation cascade (A) and on platelets (B) that were disclosed for some of these marine antithrombotics are represented at the end of this sub-chapter in Chart 1 for the reader convenience” (lines 245-247). This is the only explanation that Chart 1 that is one full page with 2 figures is given to the readers. As I previously mentioned, this information could be placed in a section on its own and fully explained as Marine drugs has no limit of words to submit manuscripts.
Reply: Chart 1 moved to page 13 and is now designated as Figure 9. We modified the overall presentation and include as mentioned the section of methods.
6. General preparation and English editing
The manuscript will require English editing. Few examples are listed below, although this is not an exhaustive list.
Line 15: “some advantages as a renew source of potential”
Lines 44-47: “Although over the last decades polysaccharides have been identified as the most therapeutically explored metabolites and suppliers of new antithrombotic agents, in fact, the less explored small molecules have proven to be also an excellent starting point for the development of new and orally effective drugs.”
Lines 50-53: “Herein, we perform a run-up on some major representatives and newest examples of the bioactive sulfated polysaccharides [covering the last five years (2014-2018)] and to include other molecules particularly with potential oral bioavailability.”
Reply: As suggested by the reviewer, the manuscript was revised by MDPI English editing service.
Round 2
Reviewer 4 Report
Revision of manuscript marinedrugs-448201 entitled “Antithrombotics from the Sea: Polysaccharides and Beyond”.
Dear authors,
Thank you for the great level of detail added to the manuscript and the comprehensive restructuration of the review. The text reads well and it will be a great guideline for future researchers interested in this field.
The text is really well written and the only modification to be made is a tiny typographical error in line 655 “Noteworthy,It is worth noting that”
Kind regards